# N-Type Nanocomposite Films Combining SWCNTs, Bi_2_Te_3_ Nanoplates, and Cationic Surfactant for Pn-Junction Thermoelectric Generators with Self-Generated Temperature Gradient Under Uniform Sunlight Irradiation

**DOI:** 10.3390/s24217060

**Published:** 2024-11-01

**Authors:** Koki Hoshino, Hisatoshi Yamamoto, Ryota Tamai, Takumi Nakajima, Shugo Miyake, Masayuki Takashiri

**Affiliations:** 1Department of Materials Science, Tokai University, 4-1-1 Kitakaname, Hiratsuka 259-1292, Kanagawa, Japan; 3cajm049@mail.u-tokai.ac.jp (K.H.); 3cajm057@mail.u-tokai.ac.jp (H.Y.); 4cajm042@tokai.ac.jp (R.T.); 4cajm046@tokai.ac.jp (T.N.); 2Department of Mechanical Engineering, Setsunan University, 17 Ikedanaka-machi, Neyagawa 572-8508, Osaka, Japan; shugo.miyake@setsunan.ac.jp

**Keywords:** Bi_2_Te_3_ nanoplates, SWCNT, pn-junction, photothermoelectric effect, thermoelectric generators

## Abstract

Flexible thermoelectric generators (TEGs) with pn-junction single-walled carbon nanotube (SWCNT) films on a polyimide substrate have attracted considerable attention for energy harvesting. This is because they generate electricity through the photo-thermoelectric effect by self-generated temperature gradient under uniform sunlight irradiation. To increase the performance and durability of the pn-junction TEGs, n-type films need to be improved as a priority. In this study, bismuth telluride (Bi_2_Te_3_) nanoplates synthesized by the solvothermal method were added to the n-type SWCNT films, including a cationic surfactant to form the nanocomposite films because Bi_2_Te_3_ has high n-type thermoelectric properties and high durability. The performances of the pn-junction TEGs were investigated by varying the heat treatment times. When the artificial sunlight was uniformly irradiated to the pn-junction TEGs, a stable output voltage of 0.47 mV was observed in the TEG with nanocomposite films heat-treated at 1 h. The output voltage decreased with increasing heat treatment time due to the decrease in the p-type region. The output voltage of TEG at 1 h is higher than that of the TEGs without Bi_2_Te_3_ nanoplates under the same conditions. Therefore, the addition of Bi_2_Te_3_ nanoplates was found to improve the performance of the pn-junction TEGs. These findings may aid in the development of facile and flexible optical devices, including photodetectors and hybrid devices integrating solar cells.

## 1. Introduction

Thermoelectric power generation is one of the most promising energy harvesting technologies. Electricity can be generated directly from ambient thermal energy. Furthermore, thermoelectric generators (TEGs) have no moving parts, which is advantageous for making TEGs smaller and thinner, and for long-term operation [1,2,3]. To generate electricity, a temperature gradient must be created in the TEGs as charge carriers diffuse from the hot side to the cold side, creating a potential difference in the TEGs. The most common method of heating TEGs is by contact with a heat source. However, in recent years there has been a surge of interest in power generation via the photo-thermoelectric effect, where light is absorbed by the TEG and subsequently converted to heat [4,5,6,7]. This allows TEGs to use heat sources with a wide range of wavelengths, from far infrared to visible light, greatly expanding the environment in which TEGs can be used.

Among thermoelectric materials, single-walled carbon nanotubes (SWCNTs) derive the greatest benefit from the photo-thermoelectric effect [8,9,10]. This is because SWCNTs have high optical absorption properties over a very wide range of wavelengths, from the visible to the far infrared [11,12,13]. In addition, SWCNTs exhibit relatively high Seebeck coefficient and electrical conductivity near 300 K [14,15,16,17,18,19], which is a consequence of their semiconducting properties depending on the structure characterized by the chiral index (*n*, *m*) [20,21,22].

While TEGs are promising energy harvesting devices, the biggest challenge is that TEGs require two different sources: a heat source and a cooling source. In other words, a TEG is unable to generate electricity through the preparation of a single heat or cold source. Under this circumstance, many researchers have proposed device structures that generate electricity from a single heat or cold source [23,24,25,26,27,28,29]. Yamasoto et al. proposed a new thermal power generation mechanism with no temperature gradient using a pn-junction of Ba_8_Au*_x_*Si_46-*x*_ clathrate crystals [23]. Matsushita et al. reported a sensitized thermal cell based on a dye-sensitized solar cell, which did not require a temperature gradient for the conversion of heat into electricity [24].

In our previous study, we prepared pn-junction SWCNT-TEGs and observed a stable output voltage under uniform artificial sunlight irradiation [30]. The output voltage generation mechanism was based on the formation of a temperature gradient from the center of the SWCNT film to the edge. This temperature gradient is the result of a difference in optical absorption between the pn-junction SWCNT film and the polyimide substrate. The gradient allows for the diffusion of electrons from the film center to the edge in the n-type region and the diffusion of holes from the film center to the edge in the p-type region. The next step is to increase the output voltage of the pn-junction SWCNT-TEGs with ultralong air stability. An ideal approach is to enhance the Seebeck coefficient of the SWCNT films. However, it is considerably challenging to enhance the Seebeck coefficient in n-type SWCNT films while maintaining their ultralong air stability.

In this study, we fabricate n-type nanocomposite films combining SWCNTs, bismuth telluride (Bi_2_Te_3_) nanoplates, and cationic surfactant for the pn-junction SWCNT-TEGs. Bi_2_Te_3_ is the best inorganic thermoelectric material, with high n-type thermoelectric performance near 300 K [31,32,33,34,35]. The crystal structure of Bi_2_Te_3_ is a rhombohedral tetradymite-type and is described as a hexagonal unit cell with the lattice parameters of *a*-axis = 0.4384 nm and *c*-axis = 3.045 nm [36]. Bi_2_Te_3_ crystals easily cleave along planes perpendicular to the *c*-axis, and this anisotropic characteristic makes it possible to fabricate nanostructures of Bi_2_Te_3_ with a sheet-like morphology such as nanoplates and nanoflakes. High-quality Bi_2_Te_3_ nanoplates or nanoflakes, which are two-dimensional materials, can enhance thermoelectric properties due to the low-dimensional and phonon scattering effects [37,38,39,40,41,42]. These nanoplates can be synthesized via a solution process [43,44,45,46,47]. Cationic surfactants are effective in maintaining the n-type properties of SWCNT films over time [48]. Therefore, when Bi_2_Te_3_ nanoplates and a cationic surfactant are combined with SWCNTs, high-performance n-type nanocomposite films can be produced without compromising the advantages of SWCNTs such as flexibility and high optical absorption. The completed SWCNT-TEGs are performed by heat treatments with different conditions, and their performances are measured under uniform artificial sunlight irradiation.

## 2. Materials and Methods

Figure 1 shows the fabrication processes of pn-junction TEGs. In Figure 1a, to prepare n-type nanocomposite films that combine SWCNTs, Bi_2_Te_3_ nanoplates, and a cationic surfactant, the Bi_2_Te_3_ nanoplates were fabricated using solvothermal synthesis. The methodology employed in the preparation of nanoplates in this study was based on that utilized in our previous study [41]. The system consisted of a stainless-steel autoclave with a built-in Teflon container, a hot plate with a magnetic stirrer, and heat blocks. The precursor solution and stir bar were placed in an autoclave with an internal volume of 50 cm^3^. Analytical-grade Bi_2_O_3_ (purity 99.9%, Fujifilm Wako Pure Chemical, Osaka, Japan), TeO_2_ (purity 99.9%, Kojundo Chemical Laboratory, Sakado, Japan), ethylene glycol (purity 99.5%, Fujifilm Wako Pure Chemical), polyvinylpyrrolidone (PVP) (purity 99.9%, Fujifilm Wako Pure Chemical, K30, Ms ~40,000), and sodium hydroxide (NaOH; (purity >97.0%, Fujifilm Wako Pure Chemical) were utilized in this study. The synthesis of Bi_2_Te_3_ nanoplates was conducted in the following manner: 0.4 g of the compound was dissolved in ethylene glycol (18 mL), followed by the addition of Bi_2_O_3_ (20 mM), TeO_2_ (70 mM) and 2 mL of a NaOH solution (5.0 M). The resulting precursor solution was subsequently sealed within an autoclave. The autoclave was then heated to a temperature of 473 K and maintained for a period of 4 h with stirring at a rate of 500 rpm. Subsequently, the precipitated products were cooled to a temperature of approximately 300 K. The products were obtained via centrifugation and subsequently washed on multiple occasions with distilled water and absolute ethanol. The precipitates were subjected to a drying process under vacuum at a temperature of 333 K for a period of 24 h.

Figure 1b shows the process of n-type and p-type dispersion. The n-type dispersion including the resulting Bi_2_Te_3_ nanoplates, SWCNTs, and cationic surfactant was prepared for fabricating n-type nanocomposite films. SWCNTs synthesized by the super-growth method (SG-CNTs) (ZEONANO SG101) (ZEON Co., Tokyo, Japan) were used as the starting material [49]. The concentrations of the SWCNTs, Bi_2_Te_3_ nanoplates, and dimethyldioctadecylammonium chloride (DODMAC) (Fujifilm Wako Pure Chemical, Osaka, Japan) as a cationic surfactant in the deionized water were 0.2, 0.002, and 1.0 wt%, respectively. The concentrations of SWCNTs and DODMAC were determined based on our previous study [48]. The concentration of Bi_2_Te_3_ nanoplates was determined by the preliminary experiments. In this experiment, nanocomposite films were prepared by varying the amount of Bi_2_Te_3_ nanoplates, which showed that the nanocomposite films lost their flexibility when an excessive amount of nanoplates were added. To efficiently disperse the solution, an ultrasonic homogenizer (Emerson, Branson Sonifier SFX 250, St. Louis, USA) was operated for 30 min at a dispersion amplitude of 60% (nominal value of 200 W) in an ice bath. The p-type dispersion, comprising solely of SWCNTs, was prepared for the fabrication of p-type SWCNT films. The concentration of the SWCNTs in ethanol was 0.2 wt%. The starting material and dispersion condition using an ultrasonic homogenizer were identical to those utilized in the preparation of the n-type nanocomposite films.

In Figure 1c, the pn-junction films were prepared via vacuum filtration using a membrane filter (PTFE, 90 mm diameter, 1 μm pore size; Advantec, Tokyo, Japan). The methodology employed in the preparation of pn-junction films in this study was based on that utilized in our previous study [7]. To create the n-type region, the area extending from the right edge of the filter to a distance of 45 mm from the center was masked. The n-type dispersion was then pipetted onto the unmasked region of the membrane filter. To prepare the p-type region, the area extending from the left edge of the filter to a distance of 45 mm from the center was masked. Thereafter, the p-type dispersion was deposited onto the unmasked region of the membrane filter. The interval between the depositions of each dispersion was set to 5 min. Following the formation of the pn-junction, the film was extracted from the membrane filter and subsequently subjected to a heat treatment process involving a mixture of Ar (95%) and H_2_ (5%) gasses under atmospheric pressure at 423 K, where the heat treatment time was varied from 1 to 3 h.

In Figure 1d, to complete the pn-junction TEG, the heat-treated films were sectioned to a sample measuring 40 mm × 10 mm near the center. The TEG was constructed by affixing the film to a polyimide sheet (60 mm × 30 mm) with double-sided adhesive tape.

The precise structure of the nanoplates was analyzed using high-resolution transmission electron microscopy TEM; JEOL, JEM-ARM200F, Akishima, Japan) and selected area electron diffraction (SAED) at an accelerating voltage of 200 kV. The microstructures and atomic compositions of the Bi_2_Te_3_ nanoplates were analyzed via field-emission scanning electron microscopy (FE-SEM; JEOL JSM-7100F) with electron backscattering diffraction. The surface morphologies of the n-type nanocomposite films were investigated using FE-SEM (Hitachi, S-4800, Tokyo, Japan). The phase purity and crystal structure of the nanoplates were characterized using X-ray diffraction (XRD; Rigaku, MiniFlex 600, Tokyo, Japan) with Cu-Kα radiation (*λ* = 0.154 nm with 2*θ* ranging from 10° to 80°).

The thermoelectric properties of the n-type nanocomposite films were evaluated in the in-plane direction at approximately 300 K. The Seebeck coefficient, *S*, was determined at approximately 300 K using a custom-built apparatus with an accuracy of ±5% [50]. One end of the film was affixed to a heat sink, while the other end was attached to a Peltier module (Z-MAX, FPH1-12704AC, Tokyo, Japan). Two K-type thermocouples with a diameter of 0.1 mm were affixed to the center of the thin films with a distance of 13 mm between them. The temperature difference between the thermocouples was varied from 0 to 4 K by controlling the electric current of the Peltier module using a DC power supply ((Kikusui, PAB32-2, Yokohama, Japan)), while the thermoelectric voltage was recorded at intervals of 1 K (temperature reader: KEYENCE, GR-3500, Osaka, Japan and digital multimeter: ADVANTEST, R6561, Tokyo, Japan). The Seebeck coefficient was estimated according to the *V*-*K* slope using the linear approximation. The Seebeck coefficient was measured four times for each sample, and the resulting values were averaged. The electrical conductivity, *σ*, was determined by implementing a four-point probe method (Napson, RT-70V, Tokyo, Japan), with an accuracy of ±3%. The electrical conductivity was measured four times for each sample, and the resulting values were averaged. The thermal conductivity, *κ*, was calculated using the following equation: *κ* = *DCρ*, where *D*, *C*, and *ρ* represent the thermal diffusivity, specific heat, and density, respectively. The thermal diffusivity was determined using non-contact laser spot periodic heating radiation thermometry (Bethel Co., TA33 thermowave analyzer, Ishioka, Japan) with an accuracy of ±5% [51]. The specific heat was determined using differential scanning calorimetry (Shimadzu, DSC-60 PLUS, Kyoko, Japan). The power factor, *PF*, and dimensionless figure of merit, *ZT*, which are crucial parameters for assessing thermoelectric performance, were calculated using the following equations: *PF* = *σS*^2^ and *ZT* = *σS*^2^*T*/*κ*, where *T* is the absolute temperature.

To evaluate the distribution of the electrical properties of the pn-junction TEGs, the in-plane Seebeck coefficient and electrical conductivity were measured linearly at seven positions at 5 mm intervals in the longitudinal direction of the film. The measurement procedures were the same as those described above for n-type nanocomposite films.

## 3. Results and Discussion

### 3.1. Characteristics of Bi_2_Te_3_ Nanoplates

A TEM image of a typical Bi_2_Te_3_ nanoplate is shown in Figure 2a. The Bi_2_Te_3_ nanoplates exhibited a regular hexagonal shape with lateral sizes of approximately 1 μm. The nanoplates were notably thin (less than 50 nm), which permitted the observation of an overlap between the nanoplates. The SAED pattern displayed in the inset of Figure 2a was indexed to the (00*l*) zone axis of rhombohedral Bi_2_Te_3_, thereby indicating that this nanoplate is single crystalline. As shown in Figure 2b, the high-resolution TEM (HRTEM) image shows that the lattice fringes exhibit structural uniformity with a spacing of 0.21 nm, which aligns closely with the d value of the (110) planes of rhombohedral Bi_2_Te_3_. The SEM image of the Bi_2_Te_3_ nanoplatelet is shown in Figure 2c, and elemental mapping analysis of the same area as the SEM image is shown in Figure 2d,e. The distribution of bismuth and tellurium in the nanoplates was found to be uniform. The concentrations of bismuth and tellurium were 38 at% and 62 at%, respectively, and were determined by the EDS spectrum and corresponding quantitative analysis, as shown in the Appendix A. Therefore, the atomic composition ratio of the nanoplates slightly deviated from the stoichiometric ratio (Bi: 40 at%, Te: 60 at%). Figure 2f shows the phase purity and crystal structure of the Bi_2_Te_3_ nanoplates examined via XRD analysis. The majority of peaks observed in the XRD pattern of the nanoplates were found to be consistent with the standard diffraction pattern of Bi_2_Te_3_ (JCPDS 15-0863). Even though the atomic composition ratio slightly deviated from the stoichiometric ratio, the nanoplates had a Bi_2_Te_3_ crystal structure.

### 3.2. Characteristics of Nanocomposite Films

Figure 3 shows the surface morphologies of n-type nanocomposite films with different treatment times. In Figure 3a, the untreated nanocomposite film had meandering SWCNT bundles primarily oriented within the plane of the film. The Bi_2_Te_3_ nanoplates were distributed over the entire surface of the film, and the basal surface of the nanoplates was in contact with the films, as shown in the inset of the enlarged image. In Figure 3b–d, the morphologies of the SWCNT bundles and Bi_2_Te_3_ nanoplates hardly changed, even when the heat treatment was carried out and the time was increased.

Figure 4 shows the in-plane electrical properties of the nanocomposite films as a function of heat treatment time. For visual convenience, the untreated electrical properties were plotted at a heat treatment time of 0 min. Figure 4a shows the Seebeck coefficient of the nanocomposite films with varying heat treatment times. For comparison, the inset of this figure shows the Seebeck coefficient of the cold-pressed Bi_2_Te_3_ nanoplate bulks and the untreated SWCNT/DODMAC films, as reported in previous studies [48,52]. The Seebeck coefficient of the untreated nanocomposite film was found to be –46 μV/K, which was between that of the Bi_2_Te_3_ nanoplate bulks of −144 μV/K and untreated SWCNT/DODMAC films without Bi_2_Te_3_ nanoplates of 22 μV/K. Therefore, Bi_2_Te_3_ nanoplates contribute to the n-type properties of the nanocomposite films without the heat treatment. When the heat treatment was performed and the treatment time was increased to 2 h, the Seebeck coefficient of the nanocomposite films increased to −60 μV/K. This was due to the evaporation of the residual ionized water used during the DODMAC addition. When further prolonging the heat treatment time, the Seebeck coefficient slightly decreased due to the DODMAC evaporation. Similar phenomena were observed in the heat treatment temperature dependence of the Seebeck coefficient of SWCNT/DODMAC films in our previous report [48]. The Seebeck coefficient, *S,* is expressed as Equation (1):(1)$$S=\frac{8\pi^{2}{k_{B}}^{2}}{3eh^{2}}m^{*}T{\left(\frac{\pi }{3n}\right)}^{\frac{2}{3}}$$
where *k_B_, h, m**, *T*, and *n* are Boltzmann constant, Planck constant, effective mass, absolute temperature, and carrier concentration, respectively. Therefore, the Seebeck coefficient is negatively correlated with carrier concentration. When the heat treatment time is increased from 0 to 2 h and the Seebeck coefficient is increased negatively from −46 to −60 μV/K, the carrier concentration is calculated to decrease by 33%, assuming no change in effective mass. On the other hand, when the heat treatment time is increased from 2 to 3 h and the Seebeck coefficient is decreased negatively from −60 to −57 μV/K, the carrier concentration is calculated to increase by 9%.

Figure 4b shows the electrical conductivity of the nanocomposite films varying the heat treatment time. The highest value of 22 S/cm was observed in the untreated nanocomposite film. When the heat treatment was performed and the treatment time was increased to 2 h, the electrical conductivity of the nanocomposite films decreased. With further prolonging of the heat treatment time, the electrical conductivity slightly increased. The electrical conductivity, *σ,* is expressed as Equation (2):(2)σ=enμ,
where *μ* is mobility. Therefore, the electrical conductivity is positively correlated with carrier concentration and mobility. When the heat treatment time was increased from 0 to 2 h, the electrical conductivity decreased by 21%, whereas the carrier concentration decreased by 23% and the mobility increased by 3% in the same region of treatment time. This indicates that the dominant factor in the decrease in electrical conductivity is the decrease in carrier concentration rather than the increase in mobility. On the other hand, when the heat treatment time was increased from 2 to 3 h, the electrical conductivity increased by 14%, whereas the carrier concentration and mobility increased by 9% and 5%, respectively, in the same region of treatment time. Although there is uncertainty when experimental errors are taken into account, we considered that the increase in electrical conductivity was due to both the increase in carrier concentration and mobility. In Figure 4c, the power factor of the nanocomposite films slightly increased with the increasing heat treatment time because the power factor depends on the Seebeck coefficient and the electrical conductivity. The highest power factor of 5.6 μW/(m·K^2^) was exhibited at the heat treatment times of 2 and 3 h.

Figure 5 shows the in-plane thermal conductivities of the nanocomposite films as a function of heat treatment time. In Figure 5a, for comparison, the inset of this figure shows the thermal conductivity of the cold-pressed Bi_2_Te_3_ nanoplate bulks and the SWCNT/DODMAC films, as reported in previous studies [41,48]. The measured thermal conductivity was almost constant at 1.4 W/(m·K) when the heat treatment time was from 0 to 2 h. This value was between the thermal conductivity of the Bi_2_Te_3_ nanoplate bulks of 1.9 W/(m·K) and that of SWCNT/DODMAC films without Bi_2_Te_3_ nanoplates of 0.6 W/(m·K). As the heat treatment time was further increased to 3 h, the measured thermal conductivity increased to 1.9 W/(m·K) due to the DODMAC evaporation. Figure 5b shows the electronic thermal conductivity of the nanocomposite films. The electronic thermal conductivity, *κ_e_*, is calculated from the measured electrical conductivity as shown in Figure 4b and the Weidemann–Franz law, which is expressed as Equation (3):(3)$$\kappa_{e}=L\sigma T,$$
where *L* is the Lorentz number (*L* = 2.44 × 10^−8^ W·Ω/K^2^). The electronic thermal conductivities of the nanocomposite films exhibit a range of 0.011 to 0.016 W/(m·K), and are significantly lower than those of the measured thermal conductivities. Figure 5c shows the lattice thermal conductivity of the nanocomposite films. The values and trends of the lattice thermal conductivity as a function of heat treatment time are mostly the same as those of the measured thermal conductivity. This is because the measured thermal conductivity is the sum of the electronic and lattice thermal conductivity components, and the electronic thermal conductivity is significantly lower than the measured thermal conductivity. The increase in the lattice thermal conductivity at a heat treatment time of 3 h is due to the enhancement of the phonon transport by decreasing the barrier of DODMAC layers. The dimensionless figure of merit, *ZT,* of the nanocomposite films is provided in the Appendix A. The results show that the highest *ZT* of 2.5 × 10^−4^ was exhibited at a heat treatment time of 1 h.

### 3.3. Performance of Pn-Junction Thermoelectric Generators

Figure 6 shows the spatial distribution of the electrical properties of the pn-junction TEGs fabricated by the different heat treatment times. In Figure 6a, the spatial distribution of the Seebeck coefficient of the pn-junction TEG without heat treatment had a point symmetry at the center. When the heat treatment time was increased, the position of the zero Seebeck coefficient shifted to the right. At the longest heat treatment time of 3 h, the measurement points of the zero Seebeck coefficient reached approximately 30 mm. Due to this phenomenon, the area of the n-type region in the TEGs increased, and the p-type region decreased. A possible explanation of this phenomenon is that elements of DODMAC rather than Bi_2_Te_3_ nanoplates gradually moved during the heat treatment. This is because the melting point of DODMAC at 420 K is comparable to the heat treatment temperature of 423 K [53], while the melting point of Bi_2_Te_3_ is 853 K. Figure 6b shows the spatial distribution of the electrical conductivity of the pn-junction TEGs. The electrical conductivity of all TEGs rapidly increased at the measurement point of 20 mm. Therefore, all TEGs exhibited lower electrical conductivity in the n-type region than in the p-type region.

Figure 7 shows the performances of the pn-junction TEGs under uniform artificial sunlight irradiation. The measurement procedure is shown in Figure 7a. The pn-junction TEG was placed on a glass wool sheet. Two copper wire electrodes were connected to both ends of the film, and the opposite ends of those copper wire electrodes were connected to a data logger (HIOKI, LR8432, Yokohama, Japan) to measure the output voltage. The performances of the TEGs were evaluated under irradiation at a light intensity of approximately 1000 W/m^2^ using an artificial sunlight source (SERIC, SOLAX 100 W XC-100 B, Koshigaya, Japan). The distance between the sunlight source and the TEG was 550 mm. The output voltage of the TEGs was measured using a data logger for 800 s after light exposure. The thermal distributions of the TEGs were measured using a thermography camera (OPTRIS, OPTXI40LTF20CFKT090, Berlin, Germany) for 800 s after sunlight irradiation, as shown in the Appendix A. As a result, all pn-junction TEGs showed that the temperature gradient was created from the center of the film to both edges of the film.

Figure 7b shows the generation mechanism of the output voltage of the pn-junction TEGs under uniform sunlight irradiation, which was demonstrated in our previous study [30]. Owing to the disparity in sunlight absorption between the TEG and the polyimide sheet, the TEG was heated more than the polyimide sheet, and heat flowed from the TEG to the polyimide sheet. Consequently, the temperature at the center of the TEG was higher, and the temperature closer to the polyimide film was lower. This phenomenon creates a temperature gradient within the TEG, even under uniform sunlight irradiation. Electrons diffuse from the film center to the n-type end in the n-type region, whereas holes diffuse from the film center to the p-type end in the p-type region. Owing to the diffusion of electrons and holes in opposite directions, an electric potential difference is created within the TEG, thereby generating an output voltage.

The time dependence of output voltages generated in the pn-junction TEGs is shown in Figure 7c. All TEGs showed stable voltage at approximately 200 s after uniform sunlight irradiation. The output voltage of the untreated TEG was exhibited at 0.30 mV. The highest output voltage of approximately 0.47 mV was exhibited by the TEG with a heat treatment time of 1 h. In contrast, the lowest output voltage of approximately 0.22 mV was exhibited by the TEG with a heat treatment time of 3 h. These trends show the influence of the Seebeck coefficient of n-type nanocomposite films and the spatial distribution of the Seebeck coefficient of the pn-junction TEGs. The untreated TEG exhibited a lower output voltage rather than the TEG with a heat treatment time of 1 h because the n-type nanocomposite film had the lowest Seebeck coefficient, although the spatial distribution of the Seebeck coefficient showed a point symmetry with the center. On the other hand, the TEG with a heat treatment time of 3 h exhibited the lowest output voltage because the region of the Seebeck p-type coefficient decreased, although the Seebeck coefficient of the n-type nanocomposite film was relatively high. Consequently, the highest output voltages were generated by the TEG with a heat treatment time of 1 h due to the remaining p-type regions and the high Seebeck coefficient. Therefore, both a high Seebeck coefficient and a point-symmetric spatial distribution of the Seebeck coefficient are required to generate high voltages in the pn-junction TEGs. In addition, the sample length of the TEGs is an important factor to improving the output voltage and maximum power. When the sample length is too short, the p- and n-type Seebeck coefficients do not reach their maximum values, and the output voltage of the TEG decreases. On the other hand, when the sample length is too long, the p- and n-type Seebeck coefficients reach their maximum values and the TEG shows a high output voltage, but the maximum power decreases because the resistance of the TEG increases. Therefore, the optimum sample length is the length at which the p- and n-type Seebeck coefficients reach maximum values at both ends of the film.

Here, we calculated the energy conversion efficiency of the pn-junction TEG heat-treated for 1 h, which exhibits the best result in this study. The energy conversion efficiency, *η*, is expressed as Equation (4): [54]
(4)$$\eta =\frac{T_{H}-T_{L}}{T_{H}}\frac{\sqrt{1+ZT_{ave}}-1}{\sqrt{1+ZT_{ave}}+T_{L}/T_{H}},$$
where *T_H_* and *T_L_* are the temperatures of the center and edge of the film in the TEG, respectively, and *ZT_ave_* is the dimensionless figure of merit at an average temperature (*T_ave_* = (*T_H_*+*T_L_*)/2). According to the Appendix A, the *T_H_*, *T_L_*, and *ZT_ave_* are 304.8 K, 299.6 K, and 2.5 × 10^−4^, respectively. Consequently, the energy conversion efficiency of the pn-junction TEG heat-treated for 1 h was calculated to be 6.2 × 10^−3^ %.

To investigate the effect of Bi_2_Te_3_ nanoplates in the films, we fabricated the pn-junction TEG without Bi_2_Te_3_ nanoplates. The heat treatment time was maintained at 1 h because the pn-junction TEG with Bi_2_Te_3_ nanoplates heat-treated for 1 h exhibited the highest output voltage. The performances of the pn-junction TEG without Bi_2_Te_3_ nanoplates, which are the spatial distribution of the Seebeck coefficient and the exposure time dependence of the output voltage, are shown in the Appendix A. It was confirmed that the spatial distribution of the Seebeck coefficient of the pn-junction TEG without Bi_2_Te_3_ nanoplates had a similar spatial distribution to that of the pn-junction TEG with Bi_2_Te_3_ nanoplates. The TEG showed stable voltage at approximately 200 s after uniform sunlight irradiation, and the average output voltage from the exposure time of 200 to 800 s was 0.35 mV. This value is lower than that of the TEG with Bi_2_Te_3_ nanoplates subjected to 1 h of heat treatment at 0.47 mV. Therefore, the addition of Bi_2_Te_3_ nanoplates was found to improve the performance of the pn-junction TEGs, but further improvements are needed, such as preventing the diffusion of surfactant during heat treatment.

## 4. Conclusions

To enhance the performance of flexible pn-junction TEGs using the photo-thermoelectric effect through self-generated temperature gradient under uniform sunlight irradiation, n-type films were improved by adding Bi_2_Te_3_ nanoplates in the SWCNTs and cationic surfactant of DODMAC. The single-crystalline Bi_2_Te_3_ nanoplates were synthesized by the solvothermal method. The structural and thermoelectric properties of the nanocomposite films were investigated, and the performances of the pn-junction TEGs were evaluated by varying the heat treatment time. The untreated TEGs exhibited an output voltage of 0.30 mV, and the output voltage increased to 0.47 mV at a heat treatment time of 1 h due to an increase in the thermoelectric properties of the nanocomposite films. To further increase the heat treatment time, the output voltage was decreased. This is because the spatial distribution of Seebeck coefficients of TEG lost the symmetry between n- and p-type, and the p-type region decreased due to the element of cationic surfactant diffused into the p-type region by the prolonged heat treatment. Compared to the output voltage of the TEGs with and without Bi_2_Te_3_ nanoplates, the TEGs with Bi_2_Te_3_ nanoplates had a heat treatment time of 1 h higher than that of the TEG without Bi_2_Te_3_ nanoplates. Therefore, the addition of Bi_2_Te_3_ nanoplates was found to improve the performance of the pn-junction TEGs. These findings may aid in the development of facile and flexible optical devices, including photodetectors and hybrid devices integrating solar cells. In photodetectors, the SWCNTs have a high light absorption characteristic in a very wide range of wavelengths from ultraviolet (UV) light (*λ* ≈ 300 nm) to terahertz wave (*λ* ≈ 1 mm). Therefore, pn-junction TEGs with SWCNTs have the potential to be used as UV sensors, infrared sensors, and terahertz detectors. To investigate whether the pn-junction TEGs can act as infrared sensors, we measured the output voltage of the TEG under uniform infrared radiation (*λ* = 2–4 μm) and observed the generation of output voltage. The results are shown in the Appendix A. In hybrid devices integrating solar cells, the output power can be increased by placing the pn-junction TEGs with SWCNTs under the solar cells. This is because the pn-junction TEG absorbs light with a wavelength below the bandgap of the solar cell material, which can then pass through the solar cell and be converted into electricity by photothermal conversion.

## Figures and Tables

**Figure 1 sensors-24-07060-f001:**
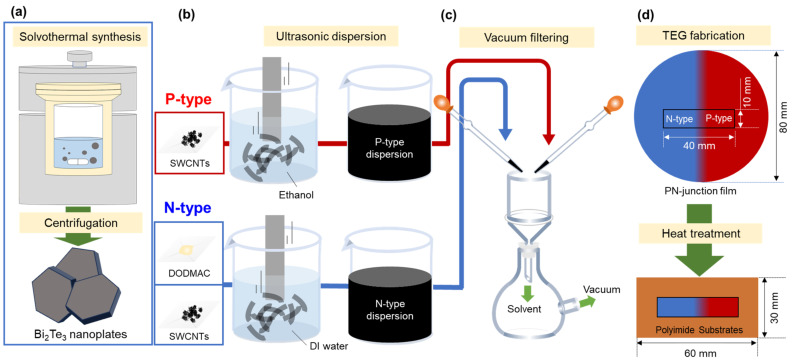
Fabrication process of pn-junction TEG. (**a**) Bi_2_Te_3_ nanoplate synthesis using a solvothermal method, (**b**) p-type and n-type dispersions using an ultrasonic homogenizer, (**c**) film preparation using vacuum filtering, and (**d**) fabrication of pn-junction TEG.

**Figure 2 sensors-24-07060-f002:**
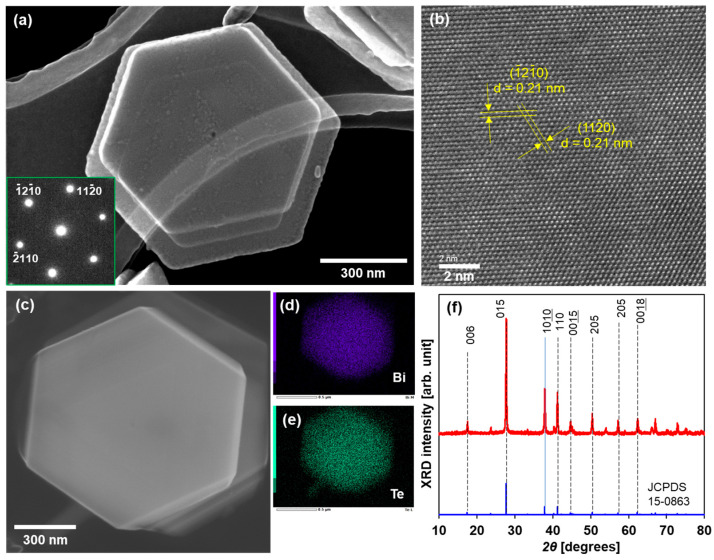
Structure of Bi_2_Te_3_ nanoplates. (**a**) TEM image and SAED pattern, (**b**) HRTEM image, (**c**) SEM image, elemental mapping of (**d**) Bi and (**e**) Te, and (**f**) XRD pattern.

**Figure 3 sensors-24-07060-f003:**
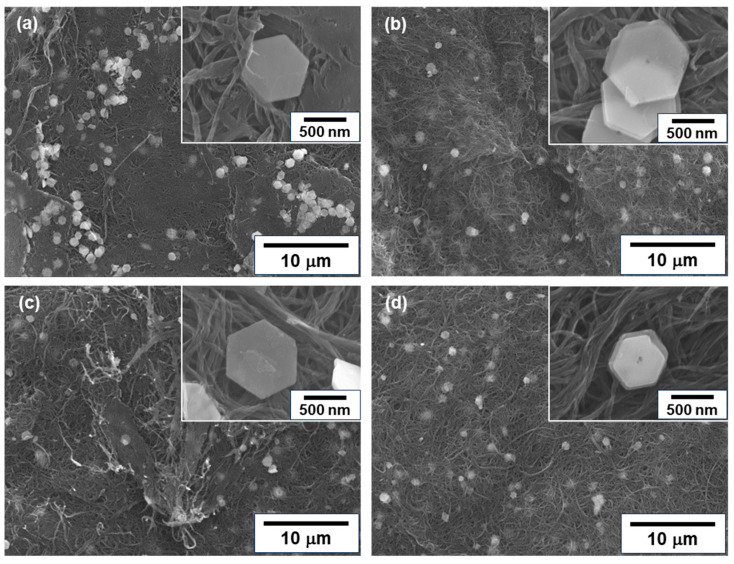
Surface morphologies of nanocomposite films analyzed using FE-SEM. (**a**) Untreated film, and films with heat treatment at (**b**) 1 h, (**c**) 2 h, and (**d**) 3 h.

**Figure 4 sensors-24-07060-f004:**
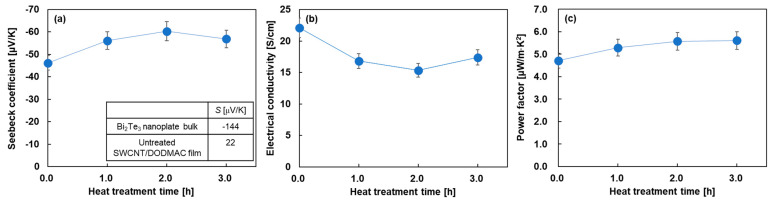
In-plane electrical properties of nanocomposite films as a function of heat treatment time. (**a**) Seebeck coefficient, (**b**) electrical conductivity, and (**c**) power factor.

**Figure 5 sensors-24-07060-f005:**
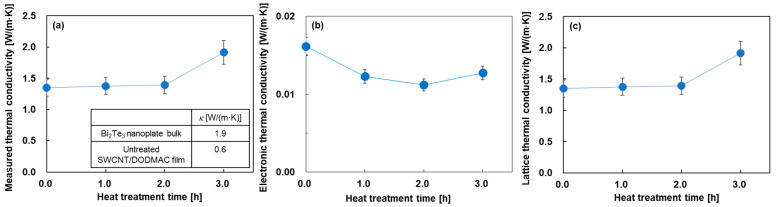
In-plane thermal properties of nanocomposite films as a function of heat treatment time. (**a**) measured thermal conductivity, (**b**) electronic thermal conductivity, and (**c**) lattice thermal conductivity.

**Figure 6 sensors-24-07060-f006:**
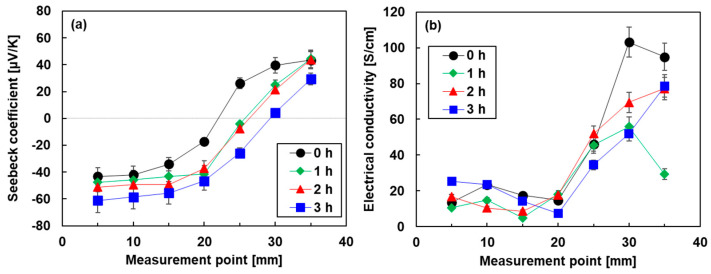
Spatial distribution of (**a**) Seebeck coefficient and (**b**) electrical conductivity of pn-junction TEGs.

**Figure 7 sensors-24-07060-f007:**
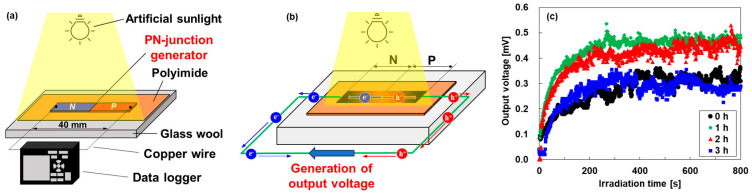
(**a**) Measurement procedure of pn-junction TEG under uniform artificial sunlight irradiation, (**b**) mechanism of generation of output voltage in TEG, and (**c**) exposure time dependence of the output voltage of TEGs.

## Data Availability

Research data can be shared by M.T. if requested.

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
