# Peer review of "N-Type Nanocomposite Films Combining SWCNTs, Bi2Te3 Nanoplates, and Cationic Surfactant for Pn-Junction Thermoelectric Generators with Self-Generated Temperature Gradient Under Uniform Sunlight Irradiation"

_sensors, 2024, doi:10.3390/s24217060_

Round 1
Reviewer 1 Report
Comments and Suggestions for Authors
Dear Authors, thanks for the well prepared manuscript and good quality data as well the data analysis has been performed by your group. I would recommend to publish the article in present form.
Author Response
Dear Reviewer:
Thank you for your valuable comments and suggestions on our manuscript.
Comment 1-1:
Dear Authors, thanks for the well prepared manuscript and good quality data as well the data analysis has been performed by your group. I would recommend to publish the article in present form.
Response 1-1:
We are very encouraged by your comment.
Reviewer 2 Report
Comments and Suggestions for Authors
In this study, bismuth telluride (Bi2Te3) nanoplates synthesized by the solvothermal method were added to the n-type SWCNT films including a cationic surfactant to form the nanocomposite films. When the artificial sunlight was uniformly irradiated to the pn-junction TEGs, the stable output voltage of 0.47 mV was observed in the TEG with nanocomposite films heat-treated at 1 h.
After reading through this study, I found this is a comprehensive study and worth publication after some minor revisions.
1. The EDS spectrum and corresponding quantitative analysis result of Figure 2 should be provided to support the chemical stoichiometry of as-prepared Bi2Te3 powders.
2. As increased carrier concentration and mobility has been proposed as the key reasons, the experimental error range should also be considered. 5-9% increase is relatively small comparing with the experimental error range, and corresponding results should be re-considered and claimed more cautiously.
3. As the key purpose of investigation in this study, the specific influence of heat treatment time should be summarized in the Abstract.
4. The fundamentals and progress of Bi2Te3-based thermoelectric materials (10.1039/d3ee02370b) should be briefly summarized to provide a better background information and deeper understanding.
5. As a key conclusion of this study is that, the addition of Bi2Te3 nano-plates was found to improve the performance of the pn-junction TEGs, the TEGs without Bi2Te3 should also be presented in performance figures for the purpose of comparison.
Author Response
Dear Reviewer:
Thank you for your valuable comments and suggestions on our manuscript. Our manuscript has been revised carefully according to the reviewer’s comments, and the detailed corrections are shown in the attached file.

Reviewer 3 Report
Comments and Suggestions for Authors
The manuscript presents a comprehensive study on the synthesis and application of n-type nanocomposite films composed of single-walled carbon nanotubes (SWCNTs) and bismuth telluride (Bi₂Te₃) nanoplates for thermoelectric generators (TEGs). The authors aim to enhance the performance of TEGs through innovative fabrication techniques and detailed characterization of the materials involved. However, the manuscript would benefit from a more comprehensive discussion of the methodology, limitations, and practical implications of the findings. Addressing these points would not only strengthen the manuscript but also provide a clearer pathway for future research in this area.
A few key issues/questions:
1. While the idea of thermoelectric generators is novel, the author didn’t mention the overall energy conversion efficiency- which is very important as this is an energy conversion device.
2. As this requires light to operate, could the authors do a more in-depth explanation on the benefit of TEGs over solar cells?
3. The author claimed it could improve facile and flexible optical devices, including photodetectors. Please explain how it would be achieved – would it be through intergration? How would the integrated device benefit from the TEG? Would the TEG compete with the other devices as they all need the photon?
4. Would the size of the sample matter for the Seebeck coefficient?
Author Response

(The authors gave the same response as above.)

Round 2
Reviewer 2 Report
Comments and Suggestions for Authors
all concerns are well addressed.
Author Response
Comment#1: All concerns are well Comments:
Response#1: We appreciate your encouraged comment.